# Machine Learning Models That Integrate Tumor Texture and Perfusion Characteristics Using Low-Dose Breast Computed Tomography Are Promising for Predicting Histological Biomarkers and Treatment Failure in Breast Cancer Patients

**DOI:** 10.3390/cancers13236013

**Published:** 2021-11-29

**Authors:** Hyun-Soo Park, Kwang-sig Lee, Bo-Kyoung Seo, Eun-Sil Kim, Kyu-Ran Cho, Ok-Hee Woo, Sung-Eun Song, Ji-Young Lee, Jaehyung Cha

**Affiliations:** 1Department of Radiology, Korea University Ansan Hospital, Korea University College of Medicine, 123 Jeokgeum-ro, Danwon-gu, Ansan-si 15355, Korea; phs9416@korea.ac.kr (H.-S.P.); eunsil1002@korea.ac.kr (E.-S.K.); dolcha@korea.ac.kr (J.C.); 2AI Center, Korea University Anam Hospital, Korea University College of Medicine, 73 Inchon-ro, Seongbuk-gu, Seoul 02841, Korea; ecophy@korea.ac.kr; 3Department of Radiology, Korea University Anam Hospital, Korea University College of Medicine, 73 Goryeodae-ro, Seongbuk-gu, Seoul 02841, Korea; krcho@korea.ac.kr (K.-R.C.); sungeuns@korea.ac.kr (S.-E.S.); 4Department of Radiology, Korea University Guro Hospital, Korea University College of Medicine, 148 Gurodong-ro, Guro-gu, Seoul 08308, Korea; wokhee@korea.ac.kr; 5Department of Radiology, Ilsan Paik Hospital, Inje University College of Medicine, 170 Juhwa-ro, Ilsanseo-gu, Goyang 10380, Korea; drleeji@paik.ac.kr; 6Cheng Hyang NF Co., Ltd., 44-5 Daehak-ro, Jongno-gu, Seoul 03122, Korea

**Keywords:** computed tomography, perfusion analysis, texture analysis, breast neoplasms, machine learning, prospective studies

## Abstract

**Simple Summary:**

Tumor angiogenesis and heterogeneity are associated with poor prognosis for breast cancer. Advances in computer technology have made it possible to noninvasively quantify tumor angiogenesis and heterogeneity appearing in imaging data. We investigated whether low-dose CT could be used as a method for functional oncology imaging to assess tumor heterogeneity and angiogenesis in breast cancer and to predict noninvasively histological biomarkers and molecular subtypes of breast cancer. Low-dose breast CT has advantages in terms of radiation safety and patient convenience. Our study produced promising results for the use of machine learning with low-dose breast CT to identify histological prognostic factors including hormone receptor and human epidermal growth factor receptor 2 status, grade, and molecular subtype in patients with invasive breast cancer. Machine learning that integrates texture and perfusion features of breast cancer with low-dose CT can provide valuable information for the realization of precision medicine.

**Abstract:**

This prospective study enrolled 147 women with invasive breast cancer who underwent low-dose breast CT (80 kVp, 25 mAs, 1.01–1.38 mSv) before treatment. From each tumor, we extracted eight perfusion parameters using the maximum slope algorithm and 36 texture parameters using the filtered histogram technique. Relationships between CT parameters and histological factors were analyzed using five machine learning algorithms. Performance was compared using the area under the receiver-operating characteristic curve (AUC) with the DeLong test. The AUCs of the machine learning models increased when using both features instead of the perfusion or texture features alone. The random forest model that integrated texture and perfusion features was the best model for prediction (AUC = 0.76). In the integrated random forest model, the AUCs for predicting human epidermal growth factor receptor 2 positivity, estrogen receptor positivity, progesterone receptor positivity, ki67 positivity, high tumor grade, and molecular subtype were 0.86, 0.76, 0.69, 0.65, 0.75, and 0.79, respectively. Entropy of pre- and postcontrast images and perfusion, time to peak, and peak enhancement intensity of hot spots are the five most important CT parameters for prediction. In conclusion, machine learning using texture and perfusion characteristics of breast cancer with low-dose CT has potential value for predicting prognostic factors and risk stratification in breast cancer patients.

## 1. Introduction

Tumor angiogenesis and heterogeneity are associated with poor prognosis for breast cancer. Angiogenesis involves rapid blood vessel formation and is necessary for the supply of oxygen and nutrients for cancer growth [1]. Abnormal angiogenesis is critical for cancer metastasis. Hematogenous metastasis occurs once the connection between the tumor and systemic circulation is established and becomes persistent [2]. Tumor heterogeneity is the one of the main causes of complicating treatment and can affect the choice of treatment strategy setting and sensitivity [3,4]. Heterogeneity has been observed between patients (intertumoral heterogeneity) and within each individual tumor (intratumoral heterogeneity) [3,4,5]. Intertumoral heterogeneity is classified according to histological biomarkers, such as hormone receptor or human epidermal growth factor receptor 2 (HER2) status, and is the basis for targeted therapy. Intratumoral heterogeneity reflects the diverse phenotypic cell populations, genetic instability, and tumor microenvironment. Therefore, assessment of tumor angiogenesis and heterogeneity in breast cancer is important for patient stratification for treatment and improving patient outcomes.

Advances in computer technology have made it possible to quantify tumor angiogenesis and heterogeneity appearing in imaging data. In breast magnetic resonance imaging (MRI), quantification of tumor angiogenesis and heterogeneity using perfusion analysis and texture analysis is effective in predicting the response to neoadjuvant chemotherapy, histological biomarkers, and molecular subtypes of breast cancer [6,7,8,9,10]. However, most MRI protocols performed before breast cancer treatment are multiparametric for the purpose of tumor characterization; these protocols include T1-weighted contrast-enhanced imaging, T2-weighted imaging, and diffusion-weighted imaging. A breast MRI scan takes 20–30 min and is complex to read because it produces more than 2500 images [11,12]. Rapid breast MRI protocols, such as ultrafast or abbreviated MRI, can shorten the scan time, but are difficult to use for assessing tumor functions and have limited use in screening high-risk populations [11,13]. The role of ultrafast or abbreviated MRI as a stand-alone examination is currently limited because of the low spatial resolution, but the assessment of tumor heterogeneity or angiogenesis properties using rapid MRI protocols is also limited.

A recent study of low-dose perfusion computed tomography (CT) has shown its feasibility for oncological functional imaging in terms of quantification of tumor angiogenesis and radiation dose [14]. That study reported correlations between the quantified CT perfusion parameters and prognostic biomarkers of breast cancer and between CT perfusion parameters and MRI enhancement characteristics [14]. Low-dose breast perfusion CT takes just 3 min. Because the perfusion protocol using low-dose CT is safe and convenient, if it can be shown to provide useful information for the treatment of breast cancer patients, it may provide an alternative for patients for whom MRI examination is difficult, for example, those who cannot undergo a long examination, have claustrophobia, or have a cardiac pacemaker. Interestingly, a recent study by Song et al. [15] showed that chest CT-based texture analysis can predict overall survival outcome in patients with inflammatory breast cancer, an aggressive malignancy. CT-based texture analysis also has been shown to be associated with survival outcomes or treatment response in a variety of solid cancers [16,17,18,19]. Therefore, we hypothesized that an integrated model to evaluate tumor angiogenesis and heterogeneity using low-dose breast CT may provide a functional tumor imaging tool for precision medicine in patients with breast cancer.

The purpose of this study was to investigate the performance of machine learning algorithms using texture and perfusion characteristics on low-dose CT to predict histological biomarkers and molecular subtypes in consecutive breast cancer patients. Recent studies have shown that applying machine learning algorithms to radiological data is useful for predicting histological factors or treatment response in breast cancer patients [6,7,20,21,22]. We evaluated CT-based predictions of histological factors and treatment failure using five supervised machine learning algorithms: logistic regression, naïve Bayes, decision tree, random forest, and artificial neural network (ANN).

## 2. Materials and Methods

### 2.1. Patients

This prospective study was approved by the Institutions Review Board of Korea University Ansan Hospital and received written informed consent from all patients. From May 2017 to October 2018, low-dose breast CT was performed in 159 consecutive patients with pathologically proven invasive breast cancer before treatment. We included patients who underwent core needle biopsy for diagnosis, had not undergone a diagnostic excision or vacuum-assisted biopsy, had no ipsilateral breast surgery within 5 years, and had no history of allergy to the CT contrast agent. Of the 159 patients, 12 were excluded for the following reasons: microinvasive cancer or ductal carcinoma in situ in the final pathology after surgery (*n* = 7), image quality deterioration caused by movement (*n* = 2), or poor identification of cancer on the perfusion map because of its small size (*n* = 3). Finally, a total of 147 patients (all women, mean age 52 years, range 25–81 years) with invasive breast cancer were enrolled in this study. After the CT scan, 85 (58%) patients had breast-conservation surgery, 44 (30%) patients underwent modified radical mastectomy, 11 (7%) patients underwent surgery after neoadjuvant chemotherapy, and seven (5%) patients received chemotherapy. A flowchart of the study population is presented in Figure 1.

### 2.2. CT Acquisition and Analysis

Low-dose breast perfusion CT was conducted according to previous studies [14,20]. Before the CT scans, a radiologist (B.K.S.) with 21 years of experience in breast imaging had reviewed previous breast images obtained using mammography, MRI, or ultrasound, and this radiologist performed targeted ultrasound to localize the breast cancers. After the cancer was identified by ultrasound, a dot skin marker (X-spot; Beekley Medical, Bristol, CT) was attached to the skin at the cancer site. For patients with multiple synchronous tumors, the largest tumor was selected. The tumor diameter on postcontrast CT ranged from 7 mm to 109 mm (mean, 24.9 ± 15.3 mm).

We used a spectral scanner (IQon Spectral CT; Philips Health Systems, Cleveland, OH) for the CT examination. CT was performed with the patient in the prone position, similar to that used for breast MRI. An additional table pad with a rectangular hole for breast placement was placed on a standard CT table for proper spread of the breast tissue for CT scans with the patient in the prone position [14]. CT was performed at 80 kVp tube voltage, 25 mAs or 30 mAs tube current, 64 × 0.625 mm collimation, 0.5 s rotation time, 512 × 512 matrix, and 5 mm slice thickness. The perfusion scan range was 40 mm along the *z*-axis, including the skin markings of the cancer. Precontrast scans were performed to determine the scan range. Once the extent of perfusion was determined, the skin markers were removed, 60 mL of the contrast agent (Xenetix 350; Guerbet, Aulnay-sous-Bois, France) was administered at 4mL/s, and 18 scans were performed at 3-s intervals and four scans at 30-s intervals. The CT dose index at 80 kVp and 25 mAs or 30 mAs using a 32 cm body phantom was 0.7 mGy or 0.9 mGy, respectively. The CT effective dose for each patient ranged from 1.01 to 1.38 mSv.

CT data were sent to a dedicated workstation (Extended Brilliance Workspace; Philips Health Systems), and perfusion analysis was performed using commercial software (Functional CT; Philips Health Systems). The maximum slope algorithm was used for analysis, and time–intensity curves and perfusion color maps for the cancers were calculated automatically when drawing the regions of interest (ROIs). A perfusion map of the cancer was obtained by (a) manually selecting images between the start and end of enhancement in the descending aorta, (b) placing an ROI on the aorta to obtain the reference arterial input curve, and (c) placing an ROI on the tumor hot spot or entire tumor extent. The size of the ROI for the hot spots was between 9.8 mm^2^ and 40.2 mm^2^. For each cancer, four perfusion parameters were measured at the tumor hot spot, and four were measured over the whole tumor. Perfusion was measured in mL/min per 100 mL, blood volume in mL/100 g, peak enhancement intensity (PEI) in Hounsfield units (HU), and time to peak (TTP) in seconds (Figure 2). Perfusion is a measure of tissue blood flow, and the volume of blood per unit time represents blood flow per unit tissue mass. The blood volume represents the total blood volume over a region during the period of the scan and is determined by the area under the time–attenuation curve. PEI indicates the peak enhancement due to contrast injection. TTP represents the time it takes to reach the peak enhancement. A hot spot is the high-perfusion area on the colored perfusion map [20]. Each perfusion parameter was measured three times, and the mean of each was used for statistical analysis.

Texture analysis was performed using CT perfusion images and a commercially available algorithm (TexRAD; Feedback Medical Ltd., London, UK). This software is a first-order statistical-based texture analysis method and uses a filtration histogram technique. Various texture analysis methods have been applied to the images, including statistical-, model-, and transform-based methods [17]. Statistical-based methods have been most commonly used to describe the relationship between gray-level values in the images and include first-order, second-order, and higher-order statistics. First-order statistical-based texture analysis evaluates the gray-level frequency distribution from the pixel intensity histogram in a given ROI. Second-order statistics can be based on co-occurrence matrices and higher-order statistics can be computed using neighboring grayscale difference matrices [23]. An optional image filtration step can be performed in texture analysis and the Laplacian of Gaussian filter is a commonly used advanced image filtering method that alters image intensity pattern and extracts specific structures corresponding to the width of the filter [17]. A lower filter value corresponds to a fine texture feature and a higher filter value emphasizes a coarse texture feature. TexRAD software is used for the initial filtration step using Laplacian of Gaussian (band-pass filter similar to a non-orthogonal Wavelet), which extracts and enhances features of different sizes and gray-level/intensity variations corresponding to spatial scale filters (SSFs) in radius [6,15]. Each SSF represents an equal number of millimeters of pixel scales, and our study used SSFs 0 (no filtration), 2 (2 mm, fine texture), and 5 (5 mm, coarse texture). Output from the filtration step includes the creation of a new derived filtered intensity texture map from the original conventional CT image corresponding to each SSF value (fine or coarse texture map). These filtered maps corresponding to the particular SSF value reflect enhancement (amplification) of objects of different sizes, numbers of objects, and gray-level intensity variation in the objects in relation to the background tissue/parenchyma. These are based on the relationship between adjacent/neighbouring pixel intensity values. Six texture parameters were extracted from the precontrast and postcontrast CT images for SSFs 0, 2, and 5, resulting in a total of 36 texture parameters (6 texture parameters × 3 SSFs × 2 imaging methods); the parameters recorded include mean pixel intensity (HU), standard deviation, mean of positive pixels (HU), entropy, kurtosis, and skewness (Figure 2). Postcontrast images were selected when the tumor was maximally enhanced during scanning. Mean of positive pixels represents the average attenuation value of >0 pixels, which reflects the average brightness of the positive pixel values. Entropy is a measure of the complexity or irregularity. Kurtosis refers to the sharpness or pointiness of the histogram pixel distribution. Skewness reflects the asymmetry of the histogram.

Image segmentation and analysis were performed by two radiologists (B.K.S. and J.Y.L., with 21 and 10 years of experience in breast imaging, respectively) who achieved consensus. They were blind to the clinicohistological findings. For the perfusion analysis, the ROIs were drawn manually for the entire tumor extent and hot spot for each cancer. For the texture analysis, the ROI was drawn manually for the entire tumor.

### 2.3. Clinicohistological Evaluation

We reviewed the histology reports to evaluate the prognostic biomarkers and molecular subtypes of breast cancer. Histology information was obtained from surgical specimens obtained from patients who underwent surgery without neoadjuvant chemotherapy (*n* = 129) and from biopsy specimens (*n* = 18). Estrogen receptor (ER), progesterone receptor (PR), and HER2 status, Ki67 index, and tumor grade were dichotomized for statistical analysis. Tumor grade was classified into low (grade 1 or 2) and high (grade 3) [24,25]. The Allred scoring system was used for ER and PR status, and a score > 2 was considered positive [26]. HER2 overexpression was considered positive when the membranes were classified as 3+ for HER2 immunohistochemical staining or 2+ for HER2 immunohistochemical staining and HER2 gene amplification in silver-stained in situ hybridization [27]. The Ki67 index was considered positive when the expression was >20% [28]. Molecular subtypes of breast cancer were divided into four categories: luminal A (ER+ or PR+, HER2−, and Ki67−); luminal B (ER+ or PR+, HER2−, and Ki67+, or ER+ or PR+, HER2+, and Ki67±); HER2-enriched (ER−, PR−, and HER2+); and triple-negative cancer (ER−, PR−, and HER2−) [29]. Table 1 shows the tumor characteristics of the 147 invasive breast cancers. The molecular subtypes were as follows: luminal A (44%, 65 of 147), luminal B (26%, 38 of 147), HER2-enriched (12%, 18 of 147), and triple-negative (18%, 26 of 147).

We reviewed the clinical data to identify treatment failure, which was defined as locoregional recurrence, contralateral breast recurrence, or distant metastasis after treatment was complete, except in patients for whom distant metastasis or contralateral breast cancer had been recorded at the time of diagnosis [30]. Distant metastasis or contralateral breast cancer was found in 10 patients at the time of diagnosis, and these patients were excluded in the evaluation of treatment failure. Locoregional recurrence was defined as ipsilateral chest wall failure or ipsilateral axillary or supra/subclavicular failure. Distant metastasis was defined as any failure outside the ipsilateral breast and regional nodes. Locoregional recurrence was confirmed histologically by tissue biopsy, and distant metastasis was confirmed by imaging studies or histological examination of tissue samples [30,31]. The median follow-up time was 39 months (range, 10–50 months). Treatment failure occurred in 17 patients, with locoregional recurrence (9 patients) and distant metastasis (8 patients).

### 2.4. Statistical Analysis

We compared CT parameters and dichotomized histological biomarker groups using the Mann–Whitney *U* test or *t* test. We evaluated differences in CT parameters between the four molecular subtypes using the Kruskal–Wallis test and analysis of variance. For data with significant differences identified in the above tests, we used post hoc analysis to identify different subgroups.

Next, we evaluated CT-based predictions of histological biomarkers and molecular subtypes using five supervised machine learning algorithms: logistic regression, naïve Bayes, decision tree, random forest, and artificial neural network (ANN). Logistic regression is a generalized linear model in which the dependent variable is categorical. In this model, the probability of a category for the dependent variable is a linear function of independent variables. The naïve Bayesian classifier is a classification model based on Bayes’ theorem. A decision tree has three components: (1) an intermediate node representing the test of an independent variable, (2) a branch representing a result of the test, and (3) a terminal node representing a category of the dependent variable. The quality of split of the decision tree was measured by the Gini index. A random forest creates many training sets, trains many decision trees, and makes predictions by majority vote (bootstrap aggregation). It can solve both regression and classification problems with large data sets. But its learning can be slow (depending on the parameterization) [32]. The random forest in this study had 1000 decision trees and the quality of split in the decision tree was measured as Gini index. The ANN is a network of neurons in several layers; in this study, the ANN had one input layer, two intermediate layers, and one output layer. Each of the intermediate layers has 10 neurons here. Neurons (basic units of information) in a previous layer connect with weights (numerical values of association) in the next layer (feedforward operation). The weights are then adjusted from the output layer (backpropagation operation). These processes are repeated until the performance of the ANN reaches a certain level. The number of intermediate layers can vary from one (perceptron) to 10 or even 1000 (deep neural network). Like the random forest, it can solve both regression and classification problems with large data sets but it can be slow in learning. Logistic regression, naïve Bayes, decision tree, random forest, and ANN are popular supervised learning approaches. Supervised learning models use labeled data, whereas their unsupervised counterparts use unlabeled data to find hidden patterns behind these unlabeled data [20,33]. All 147 cancers were divided into the training set (110 cancers, 75%) and the test set (37 cancers, 25%).

We considered six prediction tasks, corresponding to six radiological associations, which the CT-based machine learning algorithms aimed to differentiate: (a) ER+ vs. ER−, (b) PR+ vs. PR−, (c) HER2+ vs. HER2−, (d) Ki67+ vs. Ki67−, (e) tumor grade low (grade 1 and 2) vs. high (grade 3), and (f) luminal vs. nonluminal subtypes. Model performance was compared using the area under the receiver-operating characteristic curve (AUC). The random split and analysis were repeated 50 times, and their median AUC values were calculated for each model. The AUCs of the five models were compared using the DeLong test, and a *p* value < 0.05 was considered to be significant. We evaluated the performance of each model using perfusion features only, texture features only, and the combination of both. The importance ranking of CT parameters for prediction was derived from the best machine learning model. We also compared model performance in predicting histological factors and treatment failure using all CT parameters and the top five most important parameters. Analyses were performed by two statisticians (J.C. and K.L.) using SPSS Statistics (version 26.0, IBM Corp.) and Python 3.52 in August 2021.

## 3. Results

### 3.1. Associations between CT Parameters and Histological Biomarkers and Subtypes

Table 2 shows the associations between CT perfusion parameters and histological biomarkers and molecular subtypes. The perfusion of hot spots and the TTP of hot spots were significantly associated with all histological factors including ER, PR, and HER2 status, Ki67 index, grade, and molecular subtype (*p* ≤ 0.04) (Appendix A). Perfusion of hot spots differed significantly between luminal A cancers and HER2-enriched cancers and between luminal A cancers and triple-negative cancers, and TTP of hot spots differed significantly between luminal A cancers and HER-2 enriched cancers (*p* < 0.05 for all comparisons).

Table 3 shows the associations between CT texture parameters and histological factors. Entropy on contrast-enhanced CT images was significantly associated with all histological factors for all SSFs (*p* ≤ 0.02) (Appendix A). Entropy on precontrast CT images correlated significantly with ER, PR, and molecular subtype for all SSFs (*p* ≤ 0.01). Entropy on contrast-enhanced and precontrast CT images for all SSFs differed significantly between luminal A cancers and HER2-enriched cancers, and entropy on contrast-enhanced CT images differed between luminal A cancers and triple-negative cancers (*p* < 0.05 for all comparisons) (Appendix A).

### 3.2. Performance of the Machine Learning Models

Table 4 shows the AUC values of the five machine learning models to predict histological biomarkers and molecular subtypes using CT perfusion and texture features. We analyzed model performance using perfusion features only, texture features only, and the combination of both. For all machine learning models, the performance using the perfusion parameters of hot spots was better than that using the perfusion parameters of whole tumors. The integrated machine learning models had better predictive performance compared with those using the perfusion features only or the texture features only. The integrated random forest model using perfusion features for hot spots and texture features at SSF 0 had the best performance for predicting histological factors (AUC = 0.76). This value differed significantly from that for the decision tree model (AUC = 0.65, *p* < 0.05), but not from that for the naïve Bayes (AUC = 0.73, *p* = 0.63), logistic regression (AUC = 0.71, *p* = 0.41), and ANN (AUC = 0.66, *p* = 0.17) models.

The performance values for the integrated machine learning models using perfusion features of hot spots and textures features at SSF 0 for predicting each histological factor are shown in Table 5. The random forest model showed the best performance for predicting HER2 expression (AUC = 0.86); the AUC values were 0.61 for the decision tree, 0.75 for naïve Bayes, 0.67 for logistic regression, and 0.67 for ANN models. The random forest model also showed superior performance in predicting ER status, tumor grade, and molecular subtype compared with the other machine learning models.

### 3.3. Importance Ranking of CT Parameters for Prediction

Table 6 shows the top five most important CT parameters for predicting histological biomarkers and molecular subtypes of invasive breast cancer. The importance ranking of the CT parameters was obtained from the best performing integrated random forest model, which was built using the perfusion features of the hotspots and the texture features at SSF 0. The top five most important parameters were entropy on contrast-enhanced images, perfusion of hot spots, TTP of hot spots, PEI of hot spots, and entropy on precontrast images. Entropy on contrast-enhanced CT images was the most important parameter for prediction. It was significantly higher in ER−, HER2+, high grade, and HER2-enriched or triple-negative cancers (Table 7) (Appendix A). The second most important parameter, perfusion of hot spots, was also higher in those cancers with poor prognostic biomarkers. TTP of hot spots was shorter in ER−, HER2+, high grade, and non-luminal cancers (Appendix A).

Table 8 presents the AUCs for the integrated models using all CT parameters and the top five most important parameters for predicting histological factors and treatment failure. We selected median AUCs as a representative AUC of the integrated models. The random forest, naïve Bayes, and logistic regression models produced consistently higher AUC values than the decision tree or ANN model (*p* < 0.05).

## 4. Discussion

The aim of this study was to evaluate whether low-dose CT can be used as a method for functional oncology imaging to evaluate tumor heterogeneity and angiogenesis in breast cancer and to predict noninvasively histological biomarkers and molecular subtypes that are important in the treatment and prognosis of invasive cancer. We used a very low effective radiation dose (1.01–1.38 mSv) in this study. The average annual effective dose for natural background radiation in the United States is about 3 mSv and the effective dose for standard chest CT is 7 mSv [34,35,36]. We applied machine learning algorithms to low-dose breast CT for predicting the histological factors. Among the five machine learning models tested, the integrated random forest model was the best for the overall prediction of histological factors (median AUC = 0.76), and the random forest model was best for the prediction of HER2 expression (AUC = 0.86). These results are similar to those reported for an integrated model using texture and perfusion features of breast cancer on MRI (AUC = 0.80) in a recent study by Lee et al. [6]. In our study, the most important top five CT parameters for prediction were entropy on contrast-enhanced images, entropy on precontrast images, perfusion of hot spots, TTP of hot spots, and PEI of hot spots. The integrated random forest, naïve Bayes, and logistic regression models using the top five most important parameters as well as all CT parameters consistently outperformed the decision tree or ANN model for predicting histological factors and treatment failure. Therefore, our results suggest that integrated machine learning algorithms using perfusion and texture features of breast cancer may be feasible for predicting histological biomarkers and treatment failure in patients with breast cancer. Our results also suggest that the entropy and perfusion parameters of hot spots may be valuable for building CT-based predictive models for breast cancer.

In our study, entropy was the most important CT parameter for predicting histological factors in breast cancer. Entropy means texture irregularity or complexity in a histogram analysis. In our study, entropy on contrast-enhanced images and precontrast images increased in breast cancers with factors indicating poor prognosis, such as ER−, HER2+, high grade, and HER2-enriched or triple-negative cancers. Previous texture studies have shown associations between entropy and poor prognosis in a variety of cancers [6,17,37,38,39]. In breast cancer, most texture analyses have been performed using MRI. On T2-weighted images, high entropy is associated with breast cancers with poor prognosis [6,39]. By contrast, results for the relationship between entropy and cancer prognosis obtained from T1-weighted images vary depending on the image acquisition method. Lee et al. [6] found that high entropy on precontrast and postcontrast T1-weighted images in dynamic MRI was associated with histological factors indicating poor prognosis of breast cancer. Kim et al. [39] reported that low entropy on T1-weighted subtraction images was related to poor survival outcomes in patients with breast cancer and explained that using postcontrast T1-weighted images instead of subtraction images may yield different texture results. In CT texture analysis, intratumoral heterogeneity is related to cell density, necrosis, and angiogenesis [17]. Our data shows the possibility of CT texture analysis for predicting the prognosis and planning treatment using low-dose protocols in patients with breast cancer.

The CT perfusion values differ significantly between cancer and normal glandular or muscular tissues in murine breast cancer tissues and human breast cancers [14,40]. The perfusion or blood flow is higher and the time to peak or mean transit time is shorter in cancer tissues than in normal glandular or muscular tissues because of neovascularization and increased permeability of tumors. Perfusion studies of human breast cancer using low-dose CT have reported correlations between perfusion parameters and prognostic histological biomarkers [14,20]. Our results are consistent with previous findings [14,20]. Our measurements at hot spots within tumors showed stronger associations and predictive performance for histological factors of breast cancer than those obtained from the whole tumor. The top five most important parameters for prediction in our study include three perfusion parameters at hot spots: perfusion of hot spots, TTP of hot spots, and PEI of hot spots. Measurements at hot spots within tumors are more valuable than measurements in whole tumors for correlating histological angiogenesis in cancer [40,41]. Among perfusion parameters, perfusion of hot spots was one of the top five most important parameters for predicting histological factors of invasive breast cancers in a previous study by Park et al. [6]. Perfusion is a quantitative measure of blood flow through the vascular structure for a defined tissue or mass volume. Blood flow through the blood vessels is related to intratumoral velocity. Arteriovenous shunts, hyperpermeability, and immature microvessels increase the velocity of blood flow in cancers [42,43].

In this study, integrated machine learning models using texture and perfusion features improved the performance of histological factors for predicting breast cancer outcome compared with using perfusion features only or texture features only. The integrated random forest model using perfusion features for hot spots and texture features at SSF 0 had the best performance, and the AUC values for predicting each histological factor were 0.76 for ER status, 0.86 for HER2 status, and 0.79 for the molecular subtype of breast cancer. In previous studies of texture analysis of breast cancers, unfiltered or fine filtered texture results show better performance in predicting histological factors or prognosis than coarse filtered texture results [6,15,44]. A fine filter tends to enhance tissue parenchymal features, while a coarse filter enhances vascular features [45]. Therefore, unfiltered images or fine-filtered images may better represent the heterogeneity of the tumor parenchymal tissue itself than coarse-filtered images. Among the five integrated machine learning models, the random forest, logistic regression, and naïve Bayes models maintained their superior performance when comparing their performance in predicting ER status, HER2 status, molecular subtype, and treatment failure using all CT parameters and the top five most important CT parameters. Therefore, machine learning models that integrate tumor texture and perfusion characteristics using low-dose breast CT are promising for predicting histological factors and treatment failure in breast cancer patients.

Our findings will contribute to advances in breast CT. First, our CT results are similar to those of previous MRI studies of breast cancer in terms of the associations between texture and perfusion imaging features and histology [6,10,46,47]. In both CT and MRI studies, high entropy and high perfusion values are associated with breast cancers when compared with normal glands or benign lesions, and with cancers with histological biomarkers and molecular subtypes indicting poor prognosis compared with cancers with factors indicating better prognosis. In addition, a recent texture analysis study of contrast-enhanced spectral mammography using the same iodine-based contrast agents as CT shows that entropy is significantly associated with tumor grade and Ki67 expression in breast cancer [48]. They used entropy of a first-order statistical feature similar to our study. The texture characteristics of contrast-enhanced spectrum mammography have shown the diagnostic power and predictability of prognostic factors for breast cancer [48,49]. Forgia et al. [48] show high performance in distinguishing HER2 +/− (AUC = 0.91), ER +/− (AUC = 0.84), and Ki67 +/− (AUC = 0.85) using higher-order statistical texture features on contrast-enhanced spectral mammography. Therefore, the preliminary results of low-dose breast CT in this study are comparable to those of breast MRI and contrast-enhanced spectral mammography. Second, our results demonstrate the predictability of histological biomarkers and molecular subtypes of breast cancer using CT features. Three of the five machine learning models (random forest, logistic regression, and naïve Bayes) showed robust predictive performance with all CT parameters and with the top five most important CT parameters. In the eighth edition of the breast cancer staging system of the American Joint Committee on Cancer, the most significant change is the incorporation of histological biomarkers into the anatomic staging to create prognostic stages [50]. In addition, the response to neoadjuvant chemotherapy in breast cancer varies depending on the tumor subtype [51]. Therefore, hormone receptor status, HER2 status, and molecular subtypes may be important before, during, and after treatment. According to the Solid Tumor Response Evaluation Criteria version 1.1, CT and MRI are recommended as in-treatment evaluation methods, and ultrasound is not useful for evaluation because it is subjective and operator dependent [52]. Our results suggest that, because of its efficiency and convenience, low-dose breast CT may be an alternative for oncology imaging in patients unable to undergo MRI. CT has further advantages in the treatment of advanced breast cancers because CT can be used to evaluate the breast as well as extramammary sites including the lungs and lymph nodes [53,54,55]. CT also has other advantages in radiomics because CT data are less variable than MRI data [56]. In precision medicine, imaging plays a critical role in early diagnosis, treatment guidance, evaluation of treatment response, and assessment of the likelihood of recurrence [57,58]. Imaging technologies can represent morphological and functional information not visible to the naked eye and provide benefits for preventive or therapeutic interventions, reducing invasive procedures, reducing costs, minimizing the size effect of treatment, and improving prognosis. Oncology is the forefront of precision medicine. Prediction of histological biomarkers, cancer subtypes, response to adjuvant chemotherapy, and recurrence using noninvasive imaging techniques is crucial in precision medicine for breast cancer.

Our study has several limitations. First, it was conducted in one institution and no external validation was performed. However, we performed the random split analysis and repeated the analysis 50 times for internal validation, and we enrolled consecutive patients to reduce selection bias. In addition, because we used commercial software to measure tumor heterogeneity and angiogenesis, the results may be applied to other institutions for multicenter studies or clinical practice. Second, although we did not analyze segmentation reproducibility, to minimize the problem of lesion selection, two experienced breast radiologists drew the ROIs and achieved consensus on these. Perfusion analysis required the selection of hot spots on the perfusion maps, and the radiologist repeated the ROI plot three times and used the mean for statistical analysis in this study. For real clinical applications, automatic selection software for hot spots on the perfusion map is required to avoid subjective positioning ROI. Investigation of inter-reader variability in lesion selection and parameter calculations is needed to generalize and standardize our results in the near future. Third, the texture analysis software we used could analyze only two-dimensional scans, and this may not have revealed the three-dimensional texture features of the entire tumor. However, the texture analysis results from two-dimensional data and three-dimensional data were similar to those of a previous study [59]. Fourth, the scan range along the *z*-axis was only 40 mm, and the entire range of large cancers could not be included. For large cancers measuring > 40 mm, the center of the cancer was selected for scanning. Further development of perfusion imaging technology should expand the scanning range while maintaining low radiation doses in the future. Fifth, we used five basic machine learning algorithms in this study and did not consider different methods of enhancing, regularizing, and reducing associations between predictors. A wider variety of machine learning algorithms and optimization techniques could improve the effectiveness of models.

## 5. Conclusions

Our study produced promising results for the use of machine learning with low-dose breast CT to identify histological prognostic factors in patients with invasive breast cancer. Machine learning that integrates texture and perfusion features of breast cancer with low-dose CT can provide valuable information for the realization of precision medicine. Additional trials that include larger sample sizes are needed to validate and generalize the results.

## Figures and Tables

**Figure 1 cancers-13-06013-f001:**
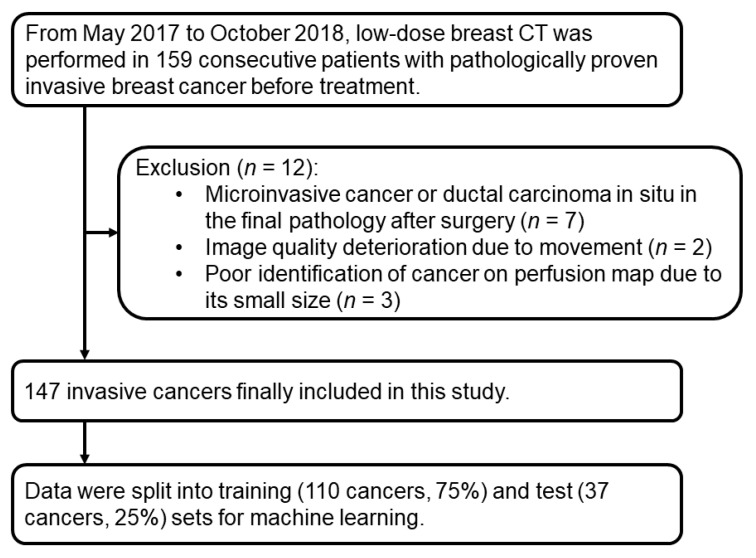
Flowchart of study population.

**Figure 2 cancers-13-06013-f002:**
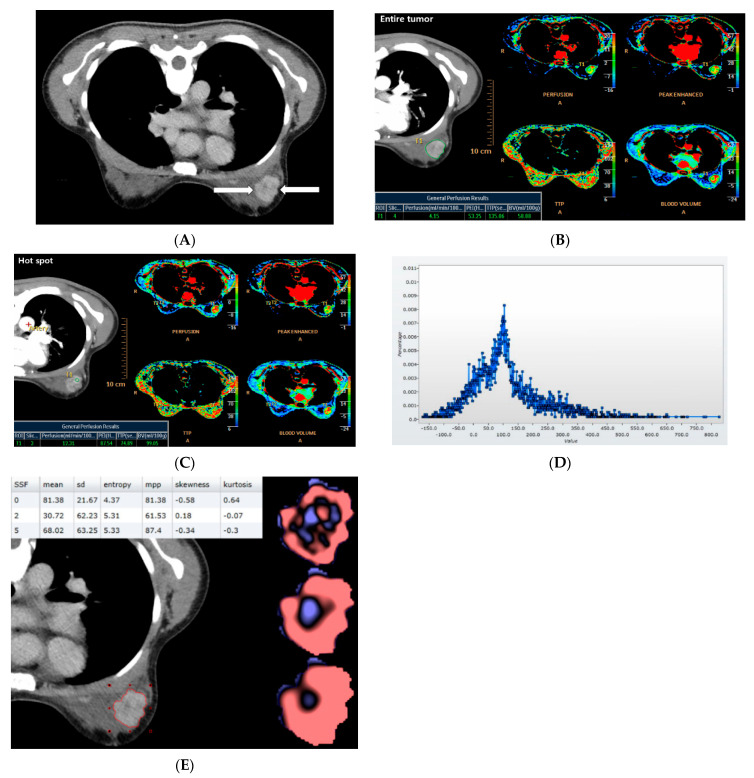
Perfusion and texture analyses on low-dose breast CT in a 47-year-old woman with invasive ductal cancer of the left breast. (**A**) Axial CT image shows an irregular shaped, irregular margined, heterogeneous enhancing mass (arrows). (**B**,**C**) Perfusion analysis was performed using the maximum slope algorithm. Regions of interest (ROIs) were drawn manually for the entire tumor extent (**B**) and hot spot (**C**) for each cancer and four perfusion parameters were measured at each ROI: perfusion, peak enhancement intensity (PEI), time to peak (TTP), and blood volume (BV). (**D**,**E**) Texture analysis was performed using a filtration histogram technique. The ROI was drawn manually for the entire tumor and a CT texture histogram was obtained (**D**). From the histogram, six statistical-based metrics were extracted (**E**). These six texture parameters were extracted from precontrast and postcontrast CT images for special scale filters (SSFs) 0, 2, and 5: mean, standard deviation (SD), entropy, mean of positive pixels (MPP), skewness, and kurtosis.

**Table 1 cancers-13-06013-t001:** Tumor characteristics.

Characteristic	Training Set (*n* = 110)	Test Set (*n* = 37)
Age	52.2 ± 9 years	52.5 ± 10 years
Tumor size	25.3 ± 16 mm	23.5 ± 14 mm
ER status		
Negative	39 (35%)	9 (24%)
Positive	71 (65%)	28 (76%)
PR status		
Negative	41 (37%)	9 (24%)
Positive	69 (63%)	28 (76%)
HER2 status		
Negative	93 (85%)	30 (81%)
Positive	17 (15%)	7 (19%)
Ki67 status		
Negative	50 (45%)	23 (62%)
Positive	60 (55%)	14 (38%)
Tumor grade		
Low	68 (62%)	28 (76%)
High	42 (38%)	9 (24%)
Molecular subtype		
Luminal	75 (68%)	28 (76%)
Non-luminal	35(32%)	9 (24%)

ER: estrogen receptor, PR: progesterone receptor, HER2: human epidermal growth factor receptor 2. Data are numbers of cancers, with percentages in parentheses.

**Table 2 cancers-13-06013-t002:** *p* values for association tests between CT perfusion parameters and histological factors in the training set.

CT Perfusion Parameter	ER	PR	HER2	Ki67	Grade	Subtype
**Hot spot**						
Perfusion	<0.001	0.001	0.01	<0.001	<0.001	<0.001
PEI	0.001	0.06	0.01	<0.001	0.001	0.001
TTP	0.001	0.02	0.04	0.02	0.01	<0.001
Blood volume	0.06	0.23	0.09	0.001	0.004	0.001
**Whole tumor**						
Perfusion	0.001	0.10	0.11	0.01	0.001	0.01
PEI	0.01	0.17	0.08	0.01	0.04	0.01
TTP	0.02	0.28	0.43	0.37	0.01	0.23
Blood volume	0.04	0.14	0.35	0.001	0.01	0.001

ER: estrogen receptor, PR: progesterone receptor, HER2: human epidermal growth factor receptor 2, PEI: peak enhancement intensity, TTP: time-to-peak enhancement. A *p* value < 0.05 for perfusion features was considered significant.

**Table 3 cancers-13-06013-t003:** *p* values for association between CT texture parameters and histological factors in the training set.

CT Texture Parameter	ER	PR	HER2	Ki67	Grade	Subtype
**SSF 0**						
Mean_precontrast	0.11	0.41	0.003	0.02	<0.001	0.01
Standard deviation_precontrast	0.55	0.18	0.15	0.07	0.30	0.01
Entropy_precontrast	0.002	<0.001	0.09	0.33	0.07	0.01
Mean of positive pixels_precontrast	0.05	0.26	0.01	0.18	<0.001	0.09
Skewness_precontrast	0.048	0.14	0.07	0.15	0.09	0.21
Kurtosis_precontrast	0.05	0.11	0.12	0.06	0.02	0.18
Mean_postccontrast	0.05	0.38	0.01	0.02	0.01	0.03
Standard deviation_postcontrast	0.56	0.41	0.16	0.24	0.51	0.05
Entropy_postcontrast	<0.001	<0.001	0.003	0.02	0.002	<0.001
Mean of positive pixels_postcontrast	0.049	0.43	0.02	0.05	0.02	0.07
Skewness_postcontrast	0.01	0.15	0.06	0.09	0.03	0.05
Kurtosis_postcontrast	0.01	0.18	0.22	0.048	0.06	0.08
**SSF 2**						
Mean_precontrast	0.046	0.12	0.12	0.01	0.06	0.01
Standard deviation_precontrast	0.82	0.19	0.19	0.92	0.68	0.32
Entropy_precontrast	<0.001	<0.001	<0.001	0.01	0.002	<0.001
Mean of positive pixels_precontrast	0.21	0.55	0.55	0.04	0.18	0.01
Skewness_precontrast	0.13	0.46	0.46	0.70	0.23	0.23
Kurtosis_precontrast	0.77	0.37	0.37	0.79	0.68	0.85
Mean_postccontrast	0.04	0.06	0.06	0.03	0.046	0.03
Standard deviation_postcontrast	0.63	0.72	0.72	0.37	0.85	0.78
Entropy_postcontrast	<0.001	<0.001	<0.001	0.001	<0.001	<0.001
Mean of positive pixels_postcontrast	0.16	0.26	0.26	0.15	0.21	0.08
Skewness_postcontrast	0.97	0.50	0.50	0.18	0.35	0.02
Kurtosis_postcontrast	0.70	0.55	0.55	0.20	0.18	0.50
**SSF 5**						
Mean_precontrast	0.18	0.29	0.21	0.02	0.27	0.04
Standard deviation_precontrast	0.88	0.55	0.10	0.84	0.86	0.79
Entropy_precontrast	<0.001	<0.001	<0.001	0.004	0.003	<0.001
Mean of positive pixels_precontrast	0.21	0.34	0.22	0.05	0.33	0.11
Skewness_precontrast	0.57	0.93	0.35	0.80	0.97	0.63
Kurtosis_precontrast	0.01	0.04	0.01	0.11	0.04	0.02
Mean_postccontrast	0.14	0.30	0.14	0.07	0.19	0.10
Standard deviation_postcontrast	0.67	0.99	0.63	0.87	0.71	0.73
Entropy_postcontrast	<0.001	<0.001	<0.001	0.001	<0.001	<0.001
Mean of positive pixels_postcontrast	0.20	0.41	0.16	0.10	0.24	0.12
Skewness_postcontrast	0.74	0.97	0.26	0.10	0.96	0.58
Kurtosis_postcontrast	0.03	0.12	0.07	0.15	0.59	0.12

ER: estrogen receptor, PR: progesterone receptor, HER2: human epidermal growth factor receptor 2, SSF: spatial scale filter. A *p* value < 0.05 for texture features was considered significant.

**Table 4 cancers-13-06013-t004:** Diagnostic performance of five machine learning models using perfusion and texture features to predict histological factors.

Machine Learning Model	Diagnostic Performance	Perfusion Features *	Texture Features ^†^	Integrating Perfusion and Texture Features ^‡^	*p* Value ^§^
Decision tree	AUC median	0.55	0.59	0.65	0.04
	AUC mean	0.55	0.58	0.62	
	AUC SD	0.25	0.34	0.28	
	AUC 95% CI	0.35, 0.75	0.42, 0.74	0.40, 0.84	
	accuracy	59%	66%	73%	
	sensitivity	68%	38%	51%	
	specificity	47%	64%	73%	
	NPV	42%	49%	61%	
	PPV	73%	57%	58%	
Naïve Bayes	AUC median	0.69	0.54	0.73	0.63
	AUC mean	0.69	0.59	0.71	
	AUC SD	0.31	0.18	0.32	
	AUC 95% CI	0.44, 0.94	0.51, 0.67	0.45, 0.97	
	accuracy	67%	51%	65%	
	sensitivity	80%	60%	60%	
	specificity	52%	67%	64%	
	NPV	56%	63%	78%	
	PPV	77%	69%	61%	
Logistic regression	AUC median	0.65	0.50	0.71	0.41
	AUC mean	0.63	0.53	0.70	
	AUC SD	0.29	0.42	0.32	
	AUC 95% CI	0.40, 0.86	0.34, 0.72	0.44, 0.96	
	accuracy	63%	62%	73%	
	sensitivity	80%	24%	46%	
	specificity	43%	68%	71%	
	NPV	53%	45%	64%	
	PPV	74%	25%	27%	
ANN	AUC median	0.60	0.56	0.66	0.17
	AUC mean	0.60	0.57	0.68	
	AUC SD	0.27	0.28	0.31	
	AUC 95% CI	0.38, 0.82	0.44, 0.70	0.43, 0.93	
	accuracy	61%	59%	68%	
	sensitivity	70%	39%	55%	
	specificity	43%	67%	72%	
	NPV	38%	47%	71%	
	PPV	74%	55%	61%	
Random forest	AUC median	0.65	0.61	0.76	…
	AUC mean	0.66	0.61	0.75	
	AUC SD	0.30	0.32	0.34	
	AUC 95% CI	0.42, 0.90	0.46, 0.76	0.48, 1.00	
	accuracy	65%	65%	74%	
	sensitivity	81%	27%	50%	
	specificity	36%	72%	76%	
	NPV	48%	48%	69%	
	PPV	72%	64%	70%	

ANN: artificial neural network, AUC: area under the receiver operating characteristic curve, SD: standard deviation, CI: confidence interval, NPV: negative predictive value, PPV: positive predictive value. * Perfusion features were measured for both the hot spots of the tumor and the whole tumor. ^†^ Texture features were measured at SSF 0, 2, and 5. ^‡^ The AUC of integrated model was highest when perfusion features of hot spots and texture features at SSF 0 were used. **^§^**
*p* values are for comparing the median AUC values with the random forest model among the integrating models by the DeLong test.

**Table 5 cancers-13-06013-t005:** AUCs and accuracies of integrated machine learning models using perfusion and texture features to predict each histological factor.

Machine Learning Model	Diagnostic Performance	ER	PR	HER2	Ki67	Grade	Subtype
Decision tree	AUC median	0.65	0.55	0.61	0.53	0.69	0.68
	accuracy	77%	59%	83%	52%	73%	73%
Naïve Bayes	AUC median	0.76	0.60	0.75	0.72	0.73	0.68
	accuracy	65%	59%	62%	68%	73%	65%
Logistic regression	AUC median	0.76	072	0.67	0.55	0.69	0.79
	accuracy	70%	73%	89%	49%	73%	81%
ANN	AUC median	0.66	0.60	0.67	0.66	0.73	0.75
	accuracy	68%	54%	84%	68%	68%	65%
Random forest	AUC median	0.76	0.69	0.86	0.65	0.75	0.79
	accuracy	76%	74%	92%	65%	67%	75%

ER: estrogen receptor, HER2: human epidermal growth factor receptor 2, AUC: area under the receiver operating characteristic curve, ANN: artificial neural network. Integrating machine learning model was built using perfusion features of hot spots and texture features at SSF 0.

**Table 6 cancers-13-06013-t006:** Top five important CT parameters from the integrated random forest model to predict histological biomarkers and molecular subtypes.

Rank	Important CT Parameters
1	Entropy_postcontrast
2	Perfusion_hot spot
3	TTP_hot spot
4	PEI_hot spot
5	Entropy_precontrast

TTP: time-to-peak enhancement, PEI: peak enhancement intensity. Integrated machine learning model was built using perfusion features of hot spots and texture features at SSF 0.

**Table 7 cancers-13-06013-t007:** Values of the top five important CT parameters according to histological factors in the training set.

Histological Facor	Entropy_ Postcontrast	Perfusion_ Hot Spot	TTP_ Hot Spot	PEI_ Hot Spot	Entropy_Precontrast
**ER**					
−	4.75 ± 0.03	46.98 ± 4.34	34.68 ± 5.70	82.22 ± 4.37	4.54 ± 0.04
+	4.50 ± 0.04	26.94 ± 3.39	51.92 ± 4.27	65.54 ± 3.44	4.35 ± 0.04
**HER2**					
−	4.55 ± 0.03	31.33 ± 3.00	47.63 ± 3.75	68.69 ± 3.05	4.40 ± 0.03
+	4.76 ± 0.03	48.88 ± 7.04	35.82 ± 9.33	86.61 ± 6.05	4.54 ± 0.05
**Grade**					
low	4.52 ± 0.04	27.67 ± 3.48	51.01 ± 4.38	64.33 ± 3.31	4.36 ± 0.04
high	4.70 ± 0.03	44.36 ± 4.37	37.38 ± 5.60	83.00 ± 4.51	4.51 ± 0.03
**Subtype**					
luminal A	4.48 ± 0.05	20.60 ± 2.67	53.20 ± 5.14	58.03 ± 3.69	4.35 ± 0.05
luminal B	4.54 ± 0.05	35.46 ± 6.74	54.59 ± 7.81	80.88 ± 5.84	4.35 ± 0.05
HER2-enriched	4.78 ± 0.04	59.22 ± 6.86	21.06 ± 4.48	86.98 ± 6.77	4.58 ± 0.04
Triple-negative	4.75 ± 0.04	44.71 ± 5.75	33.32 ± 6.83	76.89 ± 5.85	4.56 ± 0.05

ER: estrogen receptor, HER2: human epidermal growth factor receptor 2, PEI: peak enhancement intensity, TTP: time-to-peak enhancement.

**Table 8 cancers-13-06013-t008:** Comparison of AUCs according to number of CT parameters of integrated machine learning models to predict histological factors and treatment failure.

CT Parameter	AUC	*p* Value *
All parameters for predicting ER, HER2, and molecular subtype		
Decision tree	0.59	0.002
Naïve Bayes	0.75	0.52
Logistic regression	0.76	0.63
ANN	0.67	0.04
Random forest	0.79	…
Top five important parameters for predicting ER, HER2, and molecular subtype		
Decision tree	0.62	0.03
Naïve Bayes	0.76	0.95
Logistic regression	0.76	0.97
ANN	0.70	0.34
Random forest	0.76	…
All parameters for predicting ER, HER2, molecular subtype, and treatment failure		
Decision tree	0.52	<0.001
Naïve Bayes	0.81	0.29
Logistic regression	0.69	0.26
ANN	0.70	0.36
Random forest	0.76	…
Top five parameters for predicting ER, HER2, molecular subtype, and treatment failure		
Decision tree	0.52	<0.001
Naïve Bayes	0.83	0.06
Logistic regression	0.72	0.82
ANN	0.68	0.64
Random forest	0.74	…

* *p* values are for comparison with the random forest model among the integrating models by the DeLong test. ER: estrogen receptor, HER2: human epidermal growth factor receptor 2, AUC: area under the receiver operating characteristic curve, ANN: artificial neural network.

## Data Availability

The data presented in this study are available on request from the corresponding author. The data are not publicly available due to privacy.

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
