# Peer review of "Machine Learning Models That Integrate Tumor Texture and Perfusion Characteristics Using Low-Dose Breast Computed Tomography Are Promising for Predicting Histological Biomarkers and Treatment Failure in Breast Cancer Patients"

_cancers, 2021, doi:10.3390/cancers13236013_

Round 1

Reviewer 1 Report

Comment 1. As machine learning is the focus of this paper as per the paper title, Figure 1 should be expanded out to specify the algorithms that were used and the splitting, i.e., that 147 cancers were divided into the training set (110 cancers, 75%) and the test set (37 cancers, 25%) as per lines 263-264. Figure 2 also needs to be restructured and made more succinct. The legend appears to be in the middle of three pages (page 5-7), and repeats information ‘47-year-old woman’. All images should be labeled, B with ‘entire tumor’ and C with ‘hotspot’ etc. Filtration histogram technique should be better described in the text. Include more details in the histogram, e.g., 0 (no filtration), 2 (medium texture), and 5 (coarse texture). A few of these histograms (2D), from different patients, could be overlaid in a supplementary figure from low grade (panel A) and high grade (panel B) tumours to give the reader a better understanding of the variability.

Comment 2. Further details are needed on the machine learning models and statistical tests used for which dataset (e.g., lines 243-244). What software packages were used? More references are needed to detail these tests, in particular more information is needed on ANN, commenting on supervised and unsupervised learning. A brief introduction to the machine learning algorithms is included in sentences 250-263, and while it is important to include, it fits better in the introduction rather than the methods, especially given the title of the paper. As the integrated model was highest when perfusion features of hot spots and texture features at SSF 0 were used, SSF 0 should be better described and speculate why no image filtration is better. Would be useful to include more information on the accuracy with positive and negative predictive value if possible.

Further Comments:

  • Tables are hard to read and the headings of each table should state the main finding from that table where possible. Highlight in bold which are the statistically significant numbers and for table 1, 2, 3, highlight the subheadings. Table 7 needs to be colour-coded or grey-scaled. Where comparing to the random forest model, the tables should be re-titled to indicate that the p values are this comparison and the random forest model should be listed after ANN, in table 8 etc.
  • Change title to a sentence without a colon in the middle, e.g., use sentence 429-430 machine learning models that integrate tumor texture and perfusion characteristics using low-dose breast CT are promising for predicting histological factors and treatment failure in breast cancer patients
  • A total of 36 texture parameters were obtained – unclear. State that it includes pre and post contrast and with the three different filters for the six parameters specified as outlined in table 3.
  • Can the authors speculate and draw from literature about the significance that ‘entropy_postcontrast’ was the most important parameter for prediction. Too many abbreviations are used throughout, for example entropy on contrast enhanced CT images should be used rather than ‘entropy_postcontrast’ in the text of the results and discussion.

Author Response

Response to Reviewer 1 Comments

We appreciate your kind comments for our study.

We prepared all changes and additional explanation in responding to your review as follows.

Thank you so much!

===============================

Comment 1. (1) As machine learning is the focus of this paper as per the paper title, Figure 1 should be expanded out to specify the algorithms that were used and the splitting, i.e., that 147 cancers were divided into the training set (110 cancers, 75%) and the test set (37 cancers, 25%) as per lines 263-264. (2) Figure 2 also needs to be restructured and made more succinct. The legend appears to be in the middle of three pages (page 5-7), and repeats information ‘47-year-old woman’. All images should be labeled, B with ‘entire tumor’ and C with ‘hotspot’ etc. (3) Filtration histogram technique should be better described in the text. Include more details in the histogram, e.g., 0 (no filtration), 2 (medium texture), and 5 (coarse texture). A few of these histograms (2D), from different patients, could be overlaid in a supplementary figure from low grade (panel A) and high grade (panel B) tumours to give the reader a better understanding of the variability.

: Response

(1) We have revised Figure 1 as you recommended. We have added data splitting into the training and the test sets.

(2) We have revised the legend of Figure 2 as you suggested. We have deleted repeat information. We also added labels to Figure 2B and 2C with entire tumor and hot spot.

(3) We have described texture analysis methods in detail in Page 5, Line 187. In addition, we have added figures (Supplementary Figure 1) from low grade (Panel A) and high grade (Panel B) cancers to show texture characteristics as you recommended.

Comment 2. (1) Further details are needed on the machine learning models and statistical tests used for which dataset (e.g., lines 243-244). What software packages were used? (2) More references are needed to detail these tests, in particular more information is needed on ANN, commenting on supervised and unsupervised learning. (3) A brief introduction to the machine learning algorithms is included in sentences 250-263, and while it is important to include, it fits better in the introduction rather than the methods, especially given the title of the paper. (4) As the integrated model was highest when perfusion features of hot spots and texture features at SSF 0 were used, SSF 0 should be better described and speculate why no image filtration is better. (5) Would be useful to include more information on the accuracy with positive and negative predictive value if possible.

: Response

(1) We have added software packages for statistical analysis in Page 8, Line 314.

(2) We have explained in detail about machine learning in Page 7, Line 274.

(3) We have added a brief explanation of machine learning algorithms in the Introduction section in Page 3, Line 99. In addition, we have detailed machine learning methods in the Methods section in Page 7, Line 274.

(4) We have explained a possible cause of better results at SSF 0 in the Discussion section in Page 14, Line 471.

(5) We have added accuracy, sensitivity, specificity, positive predictive value, and negative predictive value in Table 4 and accuracy in Table 5.

Further Comments:

  • (1) Tables are hard to read and the headings of each table should state the main finding from that table where possible. (2) Highlight in bold which are the statistically significant numbers and for table 1, 2, 3, highlight the subheadings. (3) Table 7 needs to be colour-coded or grey-scaled. (4) Where comparing to the random forest model, the tables should be re-titled to indicate that the p values are this comparison and the random forest model should be listed after ANN, in table 8 etc.

: Response

(1) We have modified the headings of each table.

(2) We have highlighted with bold subheadings and the statistical significant numbers in Tables 1, 2, and 3. We also have highlighted with bold the statistical significant numbers in Tables 4 and 8. As recommended by Reviewer 2, we have modified Table 1 to show the tumor characteristics of the training and test sets instead of all tumors.

(3) In Table 7, we have changed horizontal and vertical entries and inserted horizontal lines for better understanding.

(4) We have listed the random forest model after ANN in Tables 4 and 8.

  • (5) Change title to a sentence without a colon in the middle, e.g., use sentence 429-430 machine learning models that integrate tumor texture and perfusion characteristics using low-dose breast CT are promising for predicting histological factors and treatment failure in breast cancer patients.

: Response

(5) We have changed the title as you mentioned.

  • (6) A total of 36 texture parameters were obtained – unclear. State that it includes pre and post contrast and with the three different filters for the six parameters specified as outlined in table 3.

: Response

(6) We have revised the description about the total number of texture parameters in Abstract in Page 1, Line 34 and in the Material and Methods section in Page 6, Line 212.

  • (7) Can the authors speculate and draw from literature about the significance that ‘entropy_postcontrast’ was the most important parameter for prediction. Too many abbreviations are used throughout, for example entropy on contrast enhanced CT images should be used rather than ‘entropy_postcontrast’ in the text of the results and discussion.

: Response

(7) We have changed ‘entropy_postcontrast’ into ‘entropy on contrast-enhanced CT images’ and ‘entropy_precontrast’ into ‘entropy on precontrast CT images’ in the text. We have also deleted abbreviations of the perfusion parameters as above.

Reviewer 2 Report

This paper describes the use of machine learning of low dose CT data to predict breast biomarkers and subtypes. The paper is reasonably well written but is not clear on several points, especially in terms of what AUC values are being presented. The authors also need to consider an initial inter-reader study to assess the reproducibility of the calculated features, the possibility that the majority of models are overparameterized, the use of alternate diagnostic metrics and some discussion of the effects of class imbalances in their data.

Points are detailed below.

Abstract

Page 1 Line 33 – This should really be 36 texture parameters, since they are calculated both pre- and post-contrast.

Page 1 Line 39 – What does this AUC value represent? The lowest AUC for the six different associations assessed? Please clarify.

Materials and Methods

Page 4 Line 147 – It would be much easier, straightforward and more robust to locate the hotspot within a pre-defined manual ROI automatically. Is there a reason for the approach taken?

Page 4 Line 147 – For the hotspot is there a pre-defined number of pixels included? This is a common approach in hotspot analysis (often the highest intensity 3x3 square). Currently, the approach taken appears very subjective.

Page 4 – Have the authors considered analyzing a subset of tumors by multiple radiologists to investigate inter-reader variations in tumor definition and calculated parameters? This is standard in current studies and would be nice to see here.

Page 7 Lines 185-193 – This description suggests that images are filtered at different scales and then parameters are determined from the calculated histograms. As such, this is not really texture analysis which looks at how pixel intensities of adjacent pixels are related to each other. Can the authors please comment as to why they regard the calculated features as texture features rather than just first order parameters of filtered images.

Page 7 Lines 200-201 – Two radiologists drew ROIs. This suggests that each radiologist did a proportion of the total number of cases. Therefore, an assessment of inter-reader variation and ICC is vital.

Page 9 Lines 258-262 – How many nodes were utilized in each hidden layer?

Page 9 Lines 263-264 – Please provide a breakdown of group sizes for the training and test sets for all radiological associations. This is important since class sizes are uneven, especially for HER2 status and molecular subtype. So, for example, how many HER2+ cases are there in the test set?

Page 9 – With such large imbalances in class size, were any forms of majority class under sampling or minority class oversampling employed? Else, models tend to classify most (if not all) cases in the majority class.

Results

Pages 9 and 10, Tables 2 and 3 – Strictly these tables should represent the results for the training cases only. Is this the case?

Page 11 Tables 4 and 5 – It is not evident what this table represents. There are six outcomes assessed but only one AUC value for each type of classifier. Do these values indicate the lowest AUCs or the average? It would be better to include all data, either here or in an appendix.

Page 12 Lines 333-341 – This seems a rather strange way to determine the most important parameters to utilize in model building. Please explain this approach, rather than the more usual techniques of employing elastic nets or maximum relevance minimum redundancy.

Page 12 Table 6 – Can the authors please comment on the fact that all the five most important features are essentially first order statistics. Both entropy parameters are calculated at SSF 0 i.e. no filtering.

Page 12 Table 7 – Again, these results should only pertain to the training set.

General – The authors need to provide information on the number of parameters used in each model. With a relatively low of number of cases it is very easy to utilize overparameterized models. Ideally there should be at least 5-10 cases in the minority class per parameter.

Page 13 Table 8 – Again, there are six outcomes but only one AUC value per classifier type here, so what exactly is been shown here?

General comment – AUC values are not particularly useful in isolation when assessing imbalanced datasets (especially HER2 status in this paper). It is more informative to utilize precision recall and metrics such as sensitivity and specificity.

Author Response

Response to Reviewer 2 Comments

We appreciate your kind comments for our study.

We prepared all changes and additional explanation in responding to your review as follows.

Thank you so much!

==================================

This paper describes the use of machine learning of low dose CT data to predict breast biomarkers and subtypes. The paper is reasonably well written but is not clear on several points, especially in terms of what AUC values are being presented. The authors also need to consider an initial inter-reader study to assess the reproducibility of the calculated features, the possibility that the majority of models are overparameterized, the use of alternate diagnostic metrics and some discussion of the effects of class imbalances in their data.

Points are detailed below.

Abstract

  1. Page 1 Line 33 – This should really be 36 texture parameters, since they are calculated both pre- and post-contrast.

: We have revised the number of texture parameters, 36.

  1. Page 1 Line 39 – What does this AUC value represent? The lowest AUC for the six different associations assessed? Please clarify.

 : We have explained about AUC in detail in Page 1, Line 40. We have added all AUC values to predict six histological factors; “In the integrated random forest model, the AUCs for predicting human epidermal growth factor receptor 2 positivity, estrogen receptor positivity, progesterone receptor positivity, ki67 positivity, high tumor grade, and molecular subtype were 0.86, 0.76, 0.69, 0.65, 0.75, and 0.79, respectively.”

Materials and Methods

  1. Page 4 Line 147 – It would be much easier, straightforward and more robust to locate the hotspot within a pre-defined manual ROI automatically. Is there a reason for the approach taken?

: For the perfusion analysis, the commercial software (Functional CT; Philips Health Systems) we used cannot segment hot spots automatically, so hand-drawn ROIs were used in this study. The hot spot is the hypervascular, high perfusion area in the color perfusion map. The hot spot does not mean the area of high intensity on CT. As our software cannot automatically segment high perfusion regions in the perfusion map, we manually drew the ROIs for the hot spots.

  1. Page 4 Line 147 – For the hotspot is there a pre-defined number of pixels included? This is a common approach in hotspot analysis (often the highest intensity 3x3 square). Currently, the approach taken appears very subjective.

: The hot spot of perfusion analysis is the hypervascular, high perfusion area in the color perfusion map. The hot spot does not mean the area of high intensity on CT. A fixed ROI is not available due to varying hot spot sizes. The size of the ROI for the hot spots was between 9.8 mm2 and 40.2 mm2 in this study. The ROI of the hot spot should not be too small in order not to select the blood vessel itself and not the high perfusion area. We have added the size of ROIs to the Materials and Methods section in Page 4, Line 156 and added limits for hand-drawn ROIs to the Discussion section in Page 15, Line 532.

  1. Page 4 – Have the authors considered analyzing a subset of tumors by multiple radiologists to investigate inter-reader variations in tumor definition and calculated parameters? This is standard in current studies and would be nice to see here.

: We did not assess inter-reader variation in selecting and calculating tumor characteristics. In this study, since two experienced breast radiologists evaluated CT characteristics in consensus, no inter-reader variation could be obtained. We have added this limitation in the Discussion section in Page 15, Line 530.

  1. Page 7 Lines 185-193 – This description suggests that images are filtered at different scales and then parameters are determined from the calculated histograms. As such, this is not really texture analysis which looks at how pixel intensities of adjacent pixels are related to each other. Can the authors please comment as to why they regard the calculated features as texture features rather than just first order parameters of filtered images.

: We have described texture analysis methods in detail in Page 5, Line 187. Many studies have been published on texture analysis of CT or MRI using the TexRADS software used in this study.

References)

  1. Chamming’s F, Ueno Y, Ferre R, et al. Features from computerized texture analysis of breast cancers at pretreatment MR imaging are associated with response to neoadjuvant chemotherapy. Radiology 2018;286(2): 412.
  2. Eun NL, Kang D, Son EJ, et al. Texture analysis with 3.0-T MRI for association of response to neoadjuvant chemotherapy in breast cancer. Radiology 2020;294:31.
  3. Lubner MG. Smith AD, Sandrasegaran K, et al. CT texture analysis: Definitions, applications, biologic correlates, and challenges. Radiographics 2017; 37:1483.
  4. Song SE, Seo BK, Cho KR, et al. Prediction of Inflammatory Breast Cancer Sur-vival Outcomes Using Computed Tomography-Based Texture Analysis. Front Bioeng Biotechnol 2021, 9, 695305.
  5. Ganeshan B, Panayiotou E, Burnand K, et al. Tumour heterogeneity in non-small cell lung carcinoma assessed by CT texture analysis: a potential marker of survival. Eur Radiol 2012;22:796.
  6. Lee JY, Lee KS, Seo BK, et al. Radiomic machine learning for predicting prognostic biomarkers and molecular subtypes of breast cancer using tumor heterogeneity and an-giogenesis properties on MRI. Eur Radiol 2021 doi: 10.1007/s00330-021-08146-8. Online ahead of print.
  7. Haider MA, Vosough A, Khalvati F, et al. CT texture analysis: a potential tool for prediction of survival in patients with metastatic clear cell carcinoma treated with sunitinib. Cancer Imaging 2017;17:4

  1. Page 7 Lines 200-201 – Two radiologists drew ROIs. This suggests that each radiologist did a proportion of the total number of cases. Therefore, an assessment of inter-reader variation and ICC is vital.

: In this study, we could not obtain inter-reader variation because two experienced breast radiologists positioned and selected ROIs in consensus.

  1. Page 9 Lines 258-262 – How many nodes were utilized in each hidden layer?

: We utilized 10 neurons in each hidden layer. We have described it in Page 8, Line 289.

  1. Page 9 Lines 263-264 – Please provide a breakdown of group sizes for the training and test sets for all radiological associations. This is important since class sizes are uneven, especially for HER2 status and molecular subtype. So, for example, how many HER2+ cases are there in the test set?

: We have modified Table 1 to show the tumor characteristics of the training and test sets instead of all tumors.

  1. Page 9 – With such large imbalances in class size, were any forms of majority class under sampling or minority class oversampling employed? Else, models tend to classify most (if not all) cases in the majority class.

: As you recommended, Table 1 was modified to show tumor characteristics from the training and test sets, and Tables 2, 3, and 7 were modified to show association test results and values using the training set rather than the entire population. The results for the entire population and training set are similar. We don’t think there is a class imbalance problem.

Results

  1. Pages 9 and 10, Tables 2 and 3 – Strictly these tables should represent the results for the training cases only. Is this the case?

: We have modified Tables 2 and 3 as the training cases. We also revised the resulting text based on the training results in Page 8, Line 319 and Page 9, Line 330.

  1. Page 11 Tables 4 and 5 – It is not evident what this table represents. There are six outcomes assessed but only one AUC value for each type of classifier. Do these values indicate the lowest AUCs or the average? It would be better to include all data, either here or in an appendix.

: In Tables 4 and 5, various diagnostic performance values and explanations have been added for better understanding.

  1. Page 12 Lines 333-341 – This seems a rather strange way to determine the most important parameters to utilize in model building. Please explain this approach, rather than the more usual techniques of employing elastic nets or maximum relevance minimum redundancy.

: We built machine learning models to evaluate predictive performance using perfusion parameters, texture parameters, and both, and then found the best performing model. We didn't build with the most important parameters in the first place.

We obtained importance ranking of CT texture and perfusion parameters because the important parameters can be applied to many different models using different metrics in different studies. We aimed that our preliminary results could expand and contribute to further research in this field. Thus, we compared predictive performance using all CT parameters and the most important parameters to evaluate usefulness of the most important CT parameters. We have added references that was done in a similar way to our study.

References)

  1. Eun NL, Kang D, Son EJ, et al. Texture Analysis with 3.0-T MRI for Association of Response to Neoadjuvant Chemotherapy in Breast Cancer. Radiology 2020;294:31
  2. Lee JY, Lee KS, Seo BK, et al. Radiomic machine learning for predicting prognostic biomarkers and molecular subtypes of breast cancer using tumor heterogeneity and angiogenesis properties on MRI. Eur Radiol 2021, doi: 10.1007/s00330-021-08146-8. Online ahead of print.

  1. Page 12 Table 6 – Can the authors please comment on the fact that all the five most important features are essentially first order statistics. Both entropy parameters are calculated at SSF 0 i.e. no filtering.

: We have described how to get the top five most important CT parameters in Page 11, Line 374. We also have modified texture analysis methods (first-order statistical methods) in detail in Page 5, Line 187.

  1. Page 12 Table 7 – Again, these results should only pertain to the training set.

: We have modified Table 7 with the training cases.

  1. General – The authors need to provide information on the number of parameters used in each model. With a relatively low of number of cases it is very easy to utilize overparameterized models. Ideally there should be at least 5-10 cases in the minority class per parameter.

: As you recommended, Table 1 was modified to show tumor characteristics from the training and test sets, and Tables 2, 3, and 7 were modified to show association test results and values using the training set rather than the entire population. The results for the entire population and training set are similar. We don’t think there is a class imbalance problem.

  1. Page 13 Table 8 – Again, there are six outcomes but only one AUC value per classifier type here, so what exactly is been shown here?

: We have added to the text how we got the AUCs in Table 8 in Page 12, Line 393. Table 8 summarizes the AUCs for the integrated models using all CT parameters and the top five most important parameters for predicting histological factors and treatment failure. We selected median AUCs as a representative AUC of the integrated models.

  1. General comment – AUC values are not particularly useful in isolation when assessing imbalanced datasets (especially HER2 status in this paper). It is more informative to utilize precision recall and metrics such as sensitivity and specificity.

: We have added accuracy, sensitivity, specificity, positive predictive value, negative predictive value in Table 4 and accuracy in Table 5.

Reviewer 3 Report

The authors used ML models and statistical analysis to correlate low-dose CT extracted features with clinical biomarkers in the case of breast cancer. The findings support the assumption that low-dose CT could be of use to predict important biomarkers, despite the low radiation and fast scanning procedure, which may delegate the captured functional information. I believe the results are publishable and the manuscript can be re-assessed after the authors consider some aspects that I mention below:

  1. The authors used baseline ML models and this is a limitation to mention and consider for the future. There are several ML algorithms that can be evaluated (e.g. SVM, Fuzzy NN, etc.) and, also, several optimization techniques that could improve the model’s effectiveness. From a technical point of view, the study lacks completeness on this front.
  2. The parameters of the decision tree, random forest, and ANN should be mentioned. For example, how many trees were used to construct the forest? How many neurons do the hidden layers of the neural network have?
  3. As far as the dataset characteristics, the evaluation methodology (50 times random train-test split), and metrics, the study is clear. However, since tests were performed for 50times, the mean and the standard deviation for the AUC scores should also be reported, besides the median.
  4. The authors could supply more Figures, especially in 3.1, 3.2, and 3.3. Figures that visualize the correlation between the CT features and histopathological factors

A minor comment:

  • In lines 210-223 the authors mention some classification and labeling criteria. Please support those selections with appropriate references.

Author Response

Response to Reviewer 3 Comments

We appreciate your kind comments for our study.

We prepared all changes and additional explanation in responding to your review as follows.

Thank you so much!

=====================================

The authors used ML models and statistical analysis to correlate low-dose CT extracted features with clinical biomarkers in the case of breast cancer. The findings support the assumption that low-dose CT could be of use to predict important biomarkers, despite the low radiation and fast scanning procedure, which may delegate the captured functional information. I believe the results are publishable and the manuscript can be re-assessed after the authors consider some aspects that I mention below:

  1. The authors used baseline ML models and this is a limitation to mention and consider for the future. There are several ML algorithms that can be evaluated (e.g. SVM, Fuzzy NN, etc.) and, also, several optimization techniques that could improve the model’s effectiveness. From a technical point of view, the study lacks completeness on this front.

: As you mentioned, we have added some limitations regarding machine learning methods to the Discussion section in Page 15, Line 545.

  1. The parameters of the decision tree, random forest, and ANN should be mentioned. For example, how many trees were used to construct the forest? How many neurons do the hidden layers of the neural network have?

: We have explained in detail about machine learning in Page 7, Line 274.

  1. As far as the dataset characteristics, the evaluation methodology (50 times random train-test split), and metrics, the study is clear. However, since tests were performed for 50times, the mean and the standard deviation for the AUC scores should also be reported, besides the median.

: We have added mean, standard deviation, and 95% confidence interval for the AUC scores in Table 4.

  1. The authors could supply more Figures, especially in 3.1, 3.2, and 3.3. Figures that visualize the correlation between the CT features and histopathological factors.

: We have added Supplementary Figure 1 to visualize the correlation between the CT texture and perfusion features and histological factors.

A minor comment:

  • In lines 210-223 the authors mention some classification and labeling criteria. Please support those selections with appropriate references.

: We have added references about each histological criterion in Page 6, Line 233.

Reviewer 4 Report

 The practical benefits of this research are not clear. Histological examination is required for the clinical application of precision medicine and no low dose CT or other radiological examination will add to its predictive value. This study has serious limitations.

Author Response

Response to Reviewer 4 Comments

We appreciate your kind comments for our study.

We prepared all changes and additional explanation in responding to your review as follows.

Thank you so much!

===========================================

 The practical benefits of this research are not clear. Histological examination is required for the clinical application of precision medicine and no low dose CT or other radiological examination will add to its predictive value. This study has serious limitations.

: We have described the value of noninvasive radiological examination in precision medicine in Page 15, Line 516.

Reviewer 5 Report

Authors should compare results not only with MRI or consider the low-dose CT a valide alternative to MRI, but also with CESM (contrast-enhanced Spectral Mammography) and with prediction signs of enhanced lesions.

Author Response

Response to Reviewer 5 Comments

We appreciate your kind comments for our study.

We prepared all changes and additional explanation in responding to your review as follows.

Thank you so much!

================================================

Authors should compare results not only with MRI or consider the low-dose CT a valid alternative to MRI, but also with CESM (contrast-enhanced Spectral Mammography) and with prediction signs of enhanced lesions.

: We have added comparison with CESM as you recommended in the Discussion section in Page 14, Line 489.

Round 2

Reviewer 2 Report

Thank you for thoroughly addressing my original comments. I have no further queries regarding this paper.

Reviewer 3 Report

The authors addressed all my comments.

Reviewer 4 Report

I have read the corrections and have no further comments.